

# Recombinant fusion protein by lysozyme and antibacterial peptide enhances ischemic wound healing via angiogenesis and reduction of inflammation in diabetic *db/db* mice

Wei Li[1,*], Yu-Xin Jiang[2,*], Qing-Ying Chen[3] and Guo-Guang Wang[1]

[1] Department of Pathophysiology, School of Basic Medicine, Wannan Medical College, Wuhu, Anhui, China
[2] Department of Basic Medicine, School of Medicine, Jiaxing University, Jiaxing, Zhejiang, China
[3] Department of Integrated traditional and western medicine, General Hospital of Jinan Military Command, Jinan, Shandong, China
* These authors contributed equally to this work.

Corresponding authors
Qing-Ying Chen, lvhx816@163.com
Guo-Guang Wang, guoguangw1226@sina.com

## ABSTRACT

**Background & aims:** Lysozyme and antibacterial peptides have been reported to broad-spectrum antibacterial activity and can further improve wound healing. The aim of this study was to assess the effectiveness of a recombinant fusion protein created by combining lysozyme and an antibacterial peptide in forming new vessels and wound healing in an ischemic hind limb.

**Methods:** An ischemic hind limb model was established by isolation and ligation of the femoral artery in diabetic *db/db* mice. Cutaneous wounds were created with or without ischemia. Adductor muscles and wounds were treated with or without the fusion protein.

**Results:** The fusion protein accelerated ischemic diabetic wound healing and attenuated impairment of ischemic adductor muscle . Further, the fusion protein elevated expression of platelet derived growth factor (PDGF) and vascular endothelial growth factor (VEGF) protein and mRNA in ischemic adductor muscle, reduced levels of tumor necrosis factor-alpha (TNF-α) and interleukin-6 (IL-6) in serum and expression of phosphorylated nuclear factor κB (p-NF-κB) and p-IKB α in ischemic adductor. The fusion protein also enhanced levels of phosphorylated VEGF and PDGF receptors in the ischemic adductor muscles from diabetic *db/db* mice.

**Conclusion:** The data showed that the beneficial effects of the fusion protein on ischemic wound healing may be associated with angiogenesis and reduction of inflammatory response in the ischemic adductor muscles of diabetic *db/db* mice.

## INTRODUCTION

Vascular disease is a severe complication of diabetes, in addition to being a major cause of mortality and morbidity in diabetic patients. Such complications can result in damage to a

number of important organs such as the heart, kidneys and retinas (*Srikanth & Deedwania, 2011*). Although the mechanisms responsible for vascular disease in diabetics remain unclear, results from numerous studies suggest that decreased expression of critical growth factors resulting from hyperglycemia can play a vital role in these diabetic vascular complications (*Schratzberger et al., 2001*). It is widely known that diabetes can cause peripheral vasculopathy and microcirculatory disturbance which contributes to ischemia in the lower limbs of patients with diabetes. Ischemia results in patients being prone to the formation of diabetic foot ulcers, exacerbating existing ulcers, and delaying diabetic wound healing (*Lobmann et al., 2002*). Thus, peripheral vasculopathy increases the risk of lower limb amputation in cases of diabetic ulcer (*Deshpande, Harris-Hayes & Schootman, 2008*). Various studies have suggested that growth factors such as vascular endothelial growth factor (VEGF) and platelet-derived growth factor (PDGF) can accelerate wound healing by increasing vessel formation (*Li et al., 2008*). In addition, clinical and experimental studies have indicated that cell transplantation can improve wound healing by accelerating the formation of blood vessels (*Amann et al., 2009*).

Lysozyme, a ubiquitous basic hydrolase in animals and plants, is an antibacterial enzyme which can kill many bacteria by hydrolyzing peptidoglycans in the bacterial cell wall (*Callewaert & Michiels, 2010*). Therefore, lysozyme is regarded as a key component of the innate immune system. It is expressed in macrophages and secreted into a number of body fluids, preventing infection from pathogenic bacteria and viruses within wounds and fractures in trauma patients (*Banks, Board & Sparks, 1986*; *Callewaert & Michiels, 2010*). Many studies have confirmed that lysozyme exhibits, as a non-specific immune factor, a variety of biological functions such as regulation of the function and proliferation of polymorphonuclear neutrophils and phagocytes, anti-tumor activity (*Royet, Gupta & Dziarski, 2011*), increased angiogenesis (*Nakatsuji & Gallo, 2012*) and wound healing, and downregulation of an inflammatory response in a number of scenarios (*Markart et al., 2004*).

Antibacterial peptides are part of a class of small molecular peptides with broad-spectrum antibacterial activity, an important defense system of a host against bacteria, fungi and some viruses (*Hilchie, Wuerth & Hancock, 2013*). These peptides are widely found in insects and mammals as important molecular barriers in the host defense against invasion from pathogenic microorganisms (*Bellm, Lehrer & Ganz, 2000*). Cecropin B is an antibacterial peptide with selective toxicity and is a broad spectrum antimicrobial which is widely found in the insects *Drosophila melanogaster* and *hyalophora* (*Seshadri Sundararajan et al., 2012*). In addition, a number of studies have indicated that some antibacterial peptides, including cecropins, possess anti-cancer activity (*Hilchie et al., 2011*), exhibit anti-inflammatory effects and promote wound healing (*Feng et al., 2012*).

Antibacterial peptides and lysozyme are easily degraded and so they are expressed at physiologically low concentrations. Therefore, to create a synergistic effect between antibacterial peptides and lysozyme and enhance their stability and expression, a fusion protein was constructed from the two through gene recombination. It has been reported

previously that fusion proteins created from antibacterial peptides and lysozyme can improve wound healing (*Koczulla & Bals, 2003*).

Individuals with diabetic vasculopathy face considerable health risks from diabetic foot ulcers and delayed wound healing. Thus, the aim of the present study was to establish a diabetic type 2 diabetes mouse model (*db/db*) with marked hyperglycemia and vascular complications to investigate the mechanisms of action of a fusion protein on angiogenesis and wound healing in the hind limbs in which ischemia has been induced.

## MATERIALS & METHODS

### Reagents and antibodies

Rabbit polyclonal antibodies towards β-actin, PDGF receptor (PDGFR) β, p-PDGFRβ, VEGF, VEGF receptor (VEGFR) 2, p-VEGFR2, nuclear factor kappa B (NF-κB), phosphorylated nuclear factor κB (p-NF-κB), inhibitor of NF-κB (IKB) α and p- IKB α were purchased from Santa Cruz Biotechnology (USA). Pentobarbital sodium and horseradish peroxidase-conjugated secondary goat anti-rabbit antibody were purchased from Sigma (Sigma Chemical Co., St. Louis, MO, USA).

### Fusion protein

The fusion protein spray was purchased from Huaibei Shangyi Biotechnology Co., Ltd (Anhui, China). The fusion protein gene was constructed from chicken lysozyme and antibacterial peptide from *Drosophila melanogaster*, and then expressed in *E. coli* BL21 (DE3).

### Animal

Forty male type 2 diabetic *db/db* mice (8–10 weeks of age) were purchased from Shanghai Laboratory Animal Co. (SLAC), Ltd (Shanghai, China), and acclimatized in the animal facility with a 12 h day/night cycle, controlled temperature of 22 ± 2 °C and 55% ± 15% humidity for two weeks prior to experimentation. This project is conducted in accordance with the National Institutes of Health Guide for the Care and Use of Laboratory Animals, and approved by Animal Experimental Ethics Committee of Wannan Medical College (20180311).

### Experimental design

Animals were randomly assigned to four groups (10 mice per group): non-ischemia control (NONISCH), non-ischemia plus administration of protein (NONISCH+PROT), ischemia control (ISCH) and ischemia plus administration of protein (ISCH+PROT). The mice were anaesthetized with 1.5% pentobarbital sodium (45 mg/kg) by intraperitoneal injection. A small incision was created and the femoral artery exposed and separated, then ligated for induction of hind limb ischemia in the corresponding group of animals. The NONISCH+PROT and ISCH+PROT mice were injected with fusion protein (0.1 mL, 50 μg/mL) in the quadriceps and gastrocnemius. A round, full-thickness skin excision (6 mm in diameter) was created dorsally. After washed with normal saline, excisions from the NONISCH+PROT and ISCH+PROT mice were sprayed with fusion protein twice per day for 14 days. The wound areas were assessed via marking on the

transparent tracing sheet along edge of the wound and analyzed by Image J software (NIH, Bethesda, MD, USA). At the end of the experiment, mice were anaesthetized with pentobarbital sodium and the carotid artery was separated for collection of blood sample via cannulation. After the mice were euthanized via bleeding, abductor muscles and granulation tissue were harvested, half of these were fixed in 4% neutral formalin for histological analysis, meanwhile the remaining tissues were stored at −80 °C until use.

## Enzyme-linked immunosorbent assay (ELISA)

Blood samples were collected to quantify proangiogenic factors in serum. The concentrations of VEGF and PDGF were determined using VEGF and PDGF-specific ELISA kits (Hefei Bomei Biotechnology Co., Ltd, Hefei, China). The concentrations of VEGF and PDGF were expressed as ng/L and ng/L, respectively.

Wound samples were homogenized in cold phosphate buffered saline (PBS) and centrifuged at 1,0000×g and 4 °C for 20 min. The supernatants were used to determine inflammatory cytokines such as tumor necrosis factor-alpha (TNF-α) and interleukin-6 (IL-6) using ELISA kits.

## Histopathology

Abductor muscles and granulation tissue were harvested and fixed in 4% formalin, then embedded in paraffin. The embedded samples were then cut into 5 μm thick sections and stained with hematoxylin and eosin (HE) to assess the inflammatory infiltrate and Masson's trichrome to measure the area of collagen within the matrix and thickness of granulation tissue.

## Immunohistochemistry

Immunohistochemistry of tissues was performed as previously describe (*Li et al., 2015*). Fixed specimens were embedded in paraffin and then cut into 5 μm thick sections. After 2 h of baking and deparaffinization, sections were treated with ethylenediamine tetraacetic acid (EDTA) buffer for antigenic retrieval, followed by incubation in hydrogen peroxide in methanol for inhibition of endogenous peroxidase. The sections were then incubated with bovine serum albumin to prevent non-specific binding. The sections were incubated with PDGF or CD34 primary antibodies (Santa Cruz Biotechnology Inc., Santa Cruz, CA, USA) overnight at 4 °C, rinsed with PBS and then incubated with biotin-conjugated secondary antibody. After washing, the sections were incubated with streptavidin-horseradish peroxidase. The sections were embedded in 3,3′-diaminobenzidine (DAB), and counterstained with hematoxylin then mounted prior to viewing.

## Real-time polymerase chain reaction (RT-PCR)

RT-PCR was performed to measure mRNA levels of VEGF, PDGF, hypoxia inducible factor (HIF)-1α, fibroblast growth factor (FGF) 2 and endothelial nitric oxide synthase (eNOS). Total RNA was extracted from the adductor muscles with Trizol reagent. mRNA was utilized to synthesize cDNA by superscript II reverse transcriptase. RT-PCR was performed by SYBR Green Real-time PCR Master Mix on an iCycler machine (Bio-Rad, Irvine, CA, USA) to amplify cDNA according to the manufacturer's instructions.

The expression of gene mRNA was normalized with the housekeeping gene β-actin. Relative changes in gene expression were analyzed using the $2^{-\Delta\Delta Ct}$ method (*Livak & Schmittgen, 2001*). Sequence primers for PCR:

VEGF (F: 5′-CACAGCAGATGTGAATGCAG-3′, R: 5′-TTTACACGTCTGCGGAT CTT-3′)

PDGF (F: 5′-CTCTTGGAGATAGACTCCGTAGG-3′, R: 5′-ACTTCTCTTCCTGC GAATGG-3′)

HIF-1α (F: 5′-GGGTACAAGAAACCACCCAT-3′, R: 5′-GAGGCTGTGTCGACTG AGAA-3′)

FGF2 (F: 5′-CAACCGGTACCTTGCTATGA-3′, R: 5′-TCCGTGACCGGTAAGTA TTG-3′)

eNOS (F: 5′- CCTTCCGCTACCAGCCAGA-3′, R: 5′-CAGAGATCTTCACTGCATT GGCTA-3′)

ACTB (F: 5′-AGTGTGACGTTGACATCCGT-3′, R: TGCTAGGAGCCAGAGCAGTA-3′).

## Western blotting

Adductor muscles were lysed in lysis buffer (50 mmol/L HEPES, 100 mmol/L sodium fluoride, 100 mmol/L sodium pyrophosphate, 1% Triton-X 100) containing 10 mmol/L sodium orthovanadate, 2 mmol/L phenyl methylsulfonyl fluoride and protease inhibitors (10 μg/L leupeptin and aprotinin). The protein in the lysate was quantified using a bicinchoninic acid (BCA) kit (Bio-Rad, Irvine, CA, USA). The lysates were electrophoretically separated using 10% SDS-PAGE and then transferred to nitrocellulose membranes. The membranes were blocked with 5% skimmed milk, and then incubated with anti β-actin, PDGFRβ, p-PDGFRβ, VEGF, VEGFR2, p-VEGFR2, NF-κB, p-NF-κB, IKB α and p- IKB α (1:1,000) antibodies, respectively. Antigens were detected and visualized using DAB.

## Statistical analysis

Values are presented as means ± standard deviation (SD) for each group. Statistical analysis was performed using an unpaired Student $t$ test or one-way analysis of variance (ANOVA) and corrected using a Bonferroni/Dunn test. $P < 0.05$ was considered statistically significant. Analysis was performed using SPSS v18.0 software (SPSS Inc., Chicago, IL, USA).

## RESULTS

### Effects of the fusion protein on ischemic wound healing

At the beginning, the wounds of every mouse were swelling and purulent, but closure of the ischemic wound was slower than that of the nonischemic wound, and the fusion protein treatment enhanced wound healing compared with ischemic wound (Fig. 1E). Wound sections stained with HE exhibited the reduction of inflammatory cell infiltration and increase of the number of blood vessels in the ischemic wound treated with the fusion protein compared with the ISCH (Figs. 1B and 1C). Hair follicles and stratified epithelium

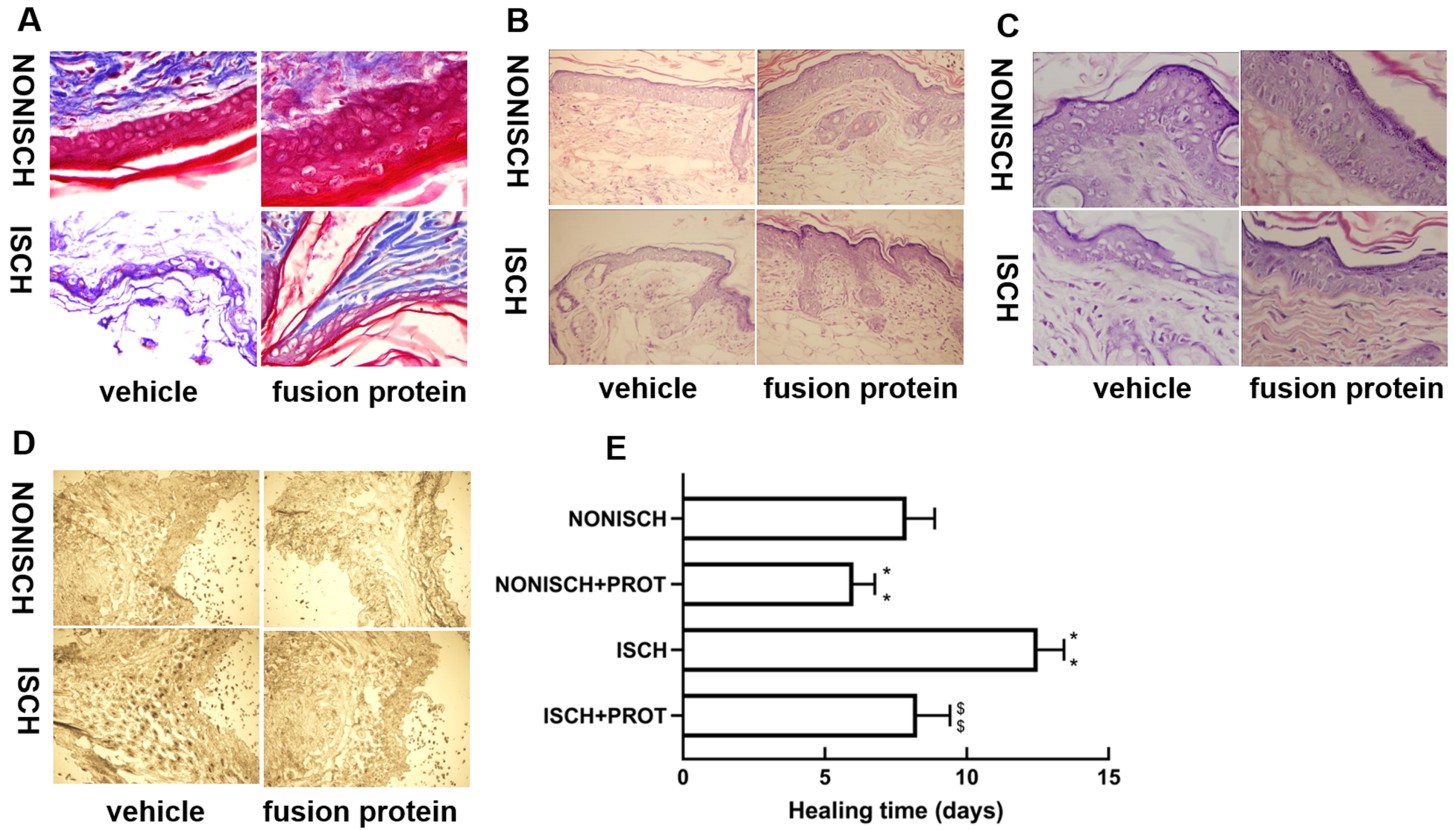

**Figure 1 Features of granulation tissue.** Representative images of granulation tissue sections. (A) Staining using Masson's trichrome of ischemic and non-ischemic wounds in diabetic mice (×400). (B) HE staining of ischemic and non-ischemic wounds in diabetic mice (×400). (C) HE staining of ischemic and non-ischemic wounds in diabetic mice (×1,000). (D) Caspase-3 staining of ischemic and non-ischemic wounds in diabetic mice (×400). Scale bar = 100 μm. (E) Duration of wound healing.                               

were observed within the wound sections of ischemic mice treated with the fusion protein (Figs. 1B and 1C).

Collagen fibers play an important role in wound healing. In this study, collagen fibers were determined with masson's trichrome staining, and the results show that collagen fibers were reduced in the ischemic diabetic wound compared with the non-ischemic wound (Fig. 1A), and collagen fibers were clearer and more regular arranged in ischemic wound treated with the fusion protein compared with the ISCH (Fig. 1A).

Furthermore, immunohistochemical staining with Caspase 3 showed that the fusion protein decreased expression of Caspase 3 (Fig. 1D)

## Effects of the fusion protein on inflammatory markers

Our results indicate that ischemia of the adductor muscle significantly increased levels of TNF-α and IL-6 compared with the NONISCH ($P < 0.05$) (Figs. 2A and 2B). Conversely, treatment with fusion protein reduced levels of TNF-α and IL-6 in the ischemic wound when compared with the ISCH group ($P < 0.05$) (Figs. 2A and 2B). But there was no significant difference in levels of TNF-α and IL-6 between the NONISCH and the NONISCH+PROT groups ($P > 0.05$) (Figs. 2A and 2B).

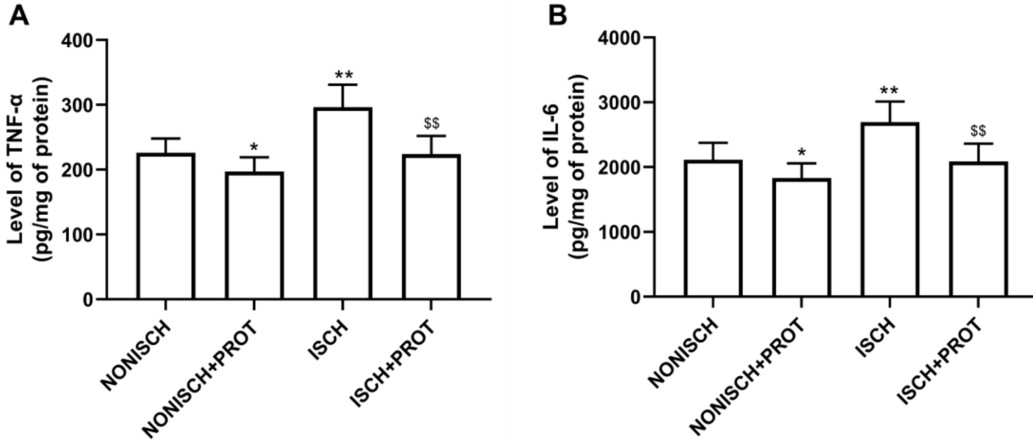

**Figure 2 Effect of the fusion protein on inflammatory cytokines in ischemic adductor muscles.** Data shown are means ± SD ($n = 8$). (A) TNF-α; (B) IL-6. *$P < 0.05$, **$P < 0.01$ vs NONISCH; $^{\$\$}P < 0.01$ vs ISCH.

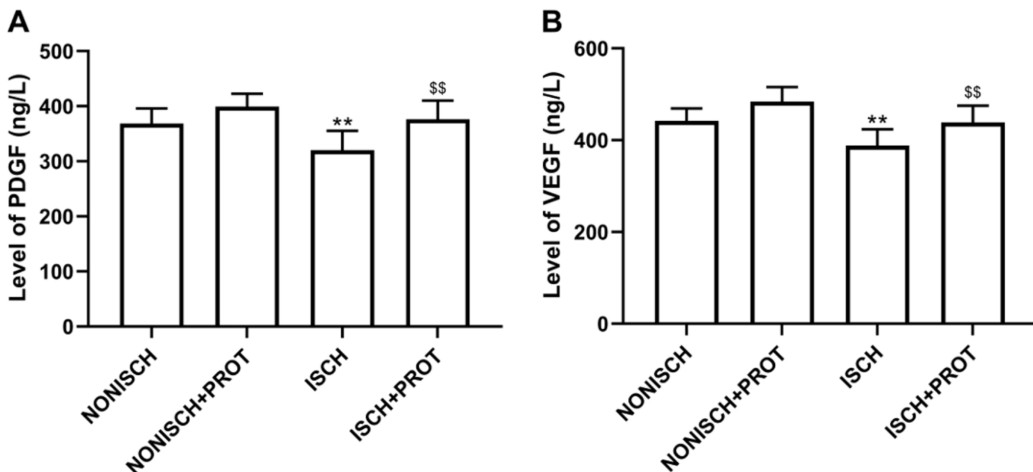

**Figure 3 Levels of proangiogenic factors in serum.** (A) Bar chart displaying PDGF concentration in plasma; (B) Bar chart displaying VEGF concentration in plasma. Data shown are means ± SD ($n = 8$). **$P < 0.01$ vs NONISCH; $^{\$\$}P < 0.01$ vs ISCH.

## Effects of the fusion protein on proangiogenic growth factor

Proangiogenic growth factors including PDGF and VEGF have been confirmed to accelerate wound closure via angiogenesis. Our results show that ischemia decreased serum levels of PDGF and VEGF compared with the NONISH group ($P < 0.05$) (Figs. 3A and 3B). However, the fusion protein elevated the levels of PDGF and VEGF in serum (**$P < 0.01$**) (Figs. 3A and 3B).

## Effects of the fusion protein on ischemic adductor muscle

Improvement of blood supply in adductor muscles plays a vital role in wound closure of ischemic lower limb. Therefore, we investigated the effect of the fusion protein on the ischemic adductor muscles. Examination of HE sections show that impaired adductor

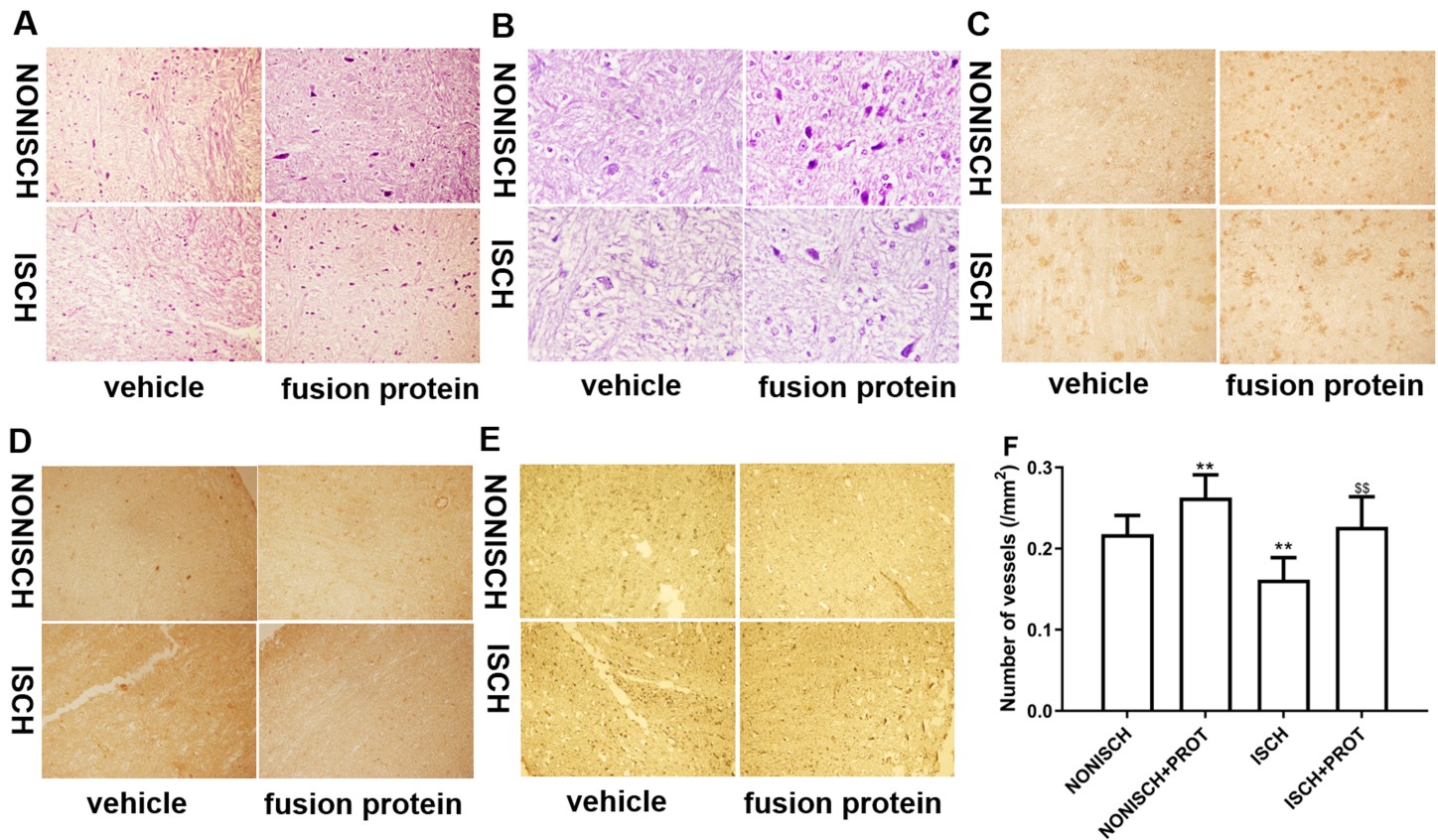

**Figure 4 Histological analysis of ischemic adductor muscles.** (A) Representative images of adductor muscle sections stained by HE (×400). (B) Representative images of adductor muscle sections stained by HE (×1,000). (C) Representative images of adductor muscle sections stained for CD34 (×400). (D) Representative images of adductor muscle sections stained for PDGF (×400). (E) Representative images of adductor muscle sections stained for Caspase 3 (×400). (F) Quantification of vascular density. Scale bar = 100 μm. **$P < 0.01$ vs NONISCH; $$P < 0.01$ vs ISCH.

muscle was observed in ischemic lower limb. Conversely, the fusion protein attenuated impairment of ischemic adductor muscle (Figs. 4A and 4B) and increased the number of capillaries compared with the ISCH group ($P < 0.05$) (Fig. 4F). Furthermore, immunohistochemical staining was used to observe proangiogenic growth factor and marker of angiogenesis. The results show that the fusion protein treatment enhanced expression of CD34 and PDGF in ischemic adductor muscle compared to the ISCH group (Figs. 4C and 4D) and reduced expression of Caspase 3 (Fig. 4E), suggesting that improvement of ischemic wound healing was associated with angiogenesis.

## Effects of the fusion protein on mRNA expression of proangiogenic growth factors

To explore improvements in wound healing due to the fusion protein in diabetic ischemic limbs, we quantified the expression of proangiogenic growth factor genes and protein. The results of RT-PCR analysis demonstrate that the expression of VEGF and PDGF mRNA decreased significantly in the adductor muscles of ischemic limbs compared with their expression in non-ischemic limbs ($P < 0.05$) (Figs. 5A and 5B). Moreover, ischemia

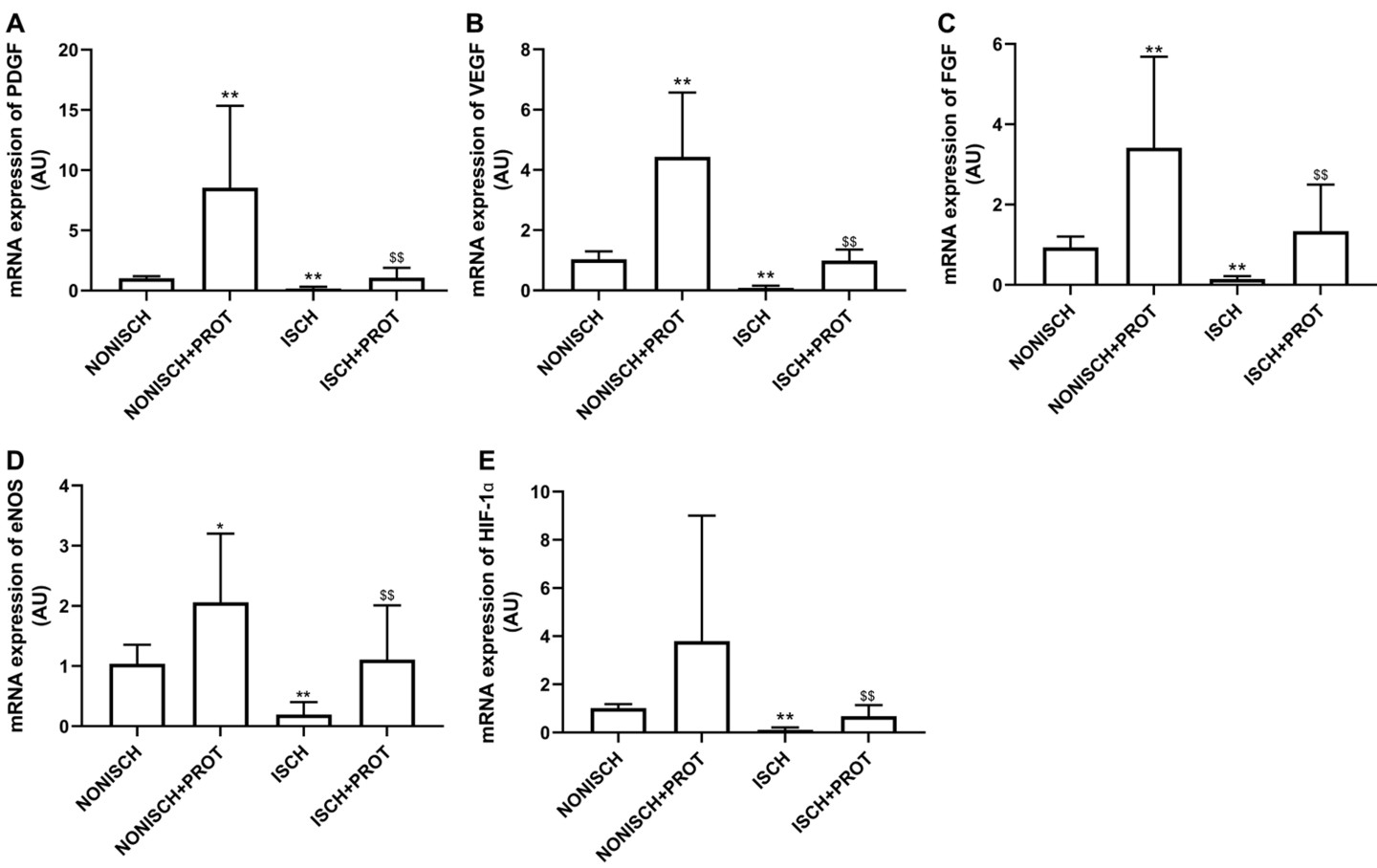

**Figure 5  mRNA expression of angiogenic factors in adductor muscles.** Bar charts displaying relative levels of mRNA of angiogenic factors. Data represent as means ± SD ($n$ = 9). β-Actin mRNA was used as the endogenous control. Relative levels of mRNA of: (A) PDGF; (B) VEGF; (C) FGF-2; (D) eNOS; (E) HIF-1a in adductor muscles of diabetic mice. *$P < 0.05$, **$P < 0.01$ vs NONISCH; $^{$$}P < 0.01$ vs ISCH.

also resulted in decreased expression of FGF-2, eNOS and HIF-1α mRNA in the adductor muscles ($P < 0.05$) (Figs. 5C–5E). Treatment using fusion protein increased expression of these growth factors in the muscles of ischemic limbs when compared with the ISCH group ($P < 0.05$) (Figs. 5A–5E).

## Effects of the fusion protein on expression of proangiogenic growth factor in ischemic adductor muscles

In this study, we further examined changes of VEGF and PDGF protein expression in the ischemic adductor muscles. The expression of VEGF and PDGF protein decreased in the adductor muscles of ischemic limbs compared with non-ischemic limbs, while ischemia resulted in decreased phosphorylation of their respective receptors VEGFR2 and PDGFR-β. Treatment with fusion protein increased phosphorylation of PDGFR-β ($P < 0.05$) (Figs. 6A and 6C) and VEGFR2 ($P < 0.05$) (Figs. 6B and 6D), and expression of PDGF ($P < 0.05$) (Fig. 6E) and VEGF proteins compared with the ISCH group ($P < 0.05$) (Fig. 6F).

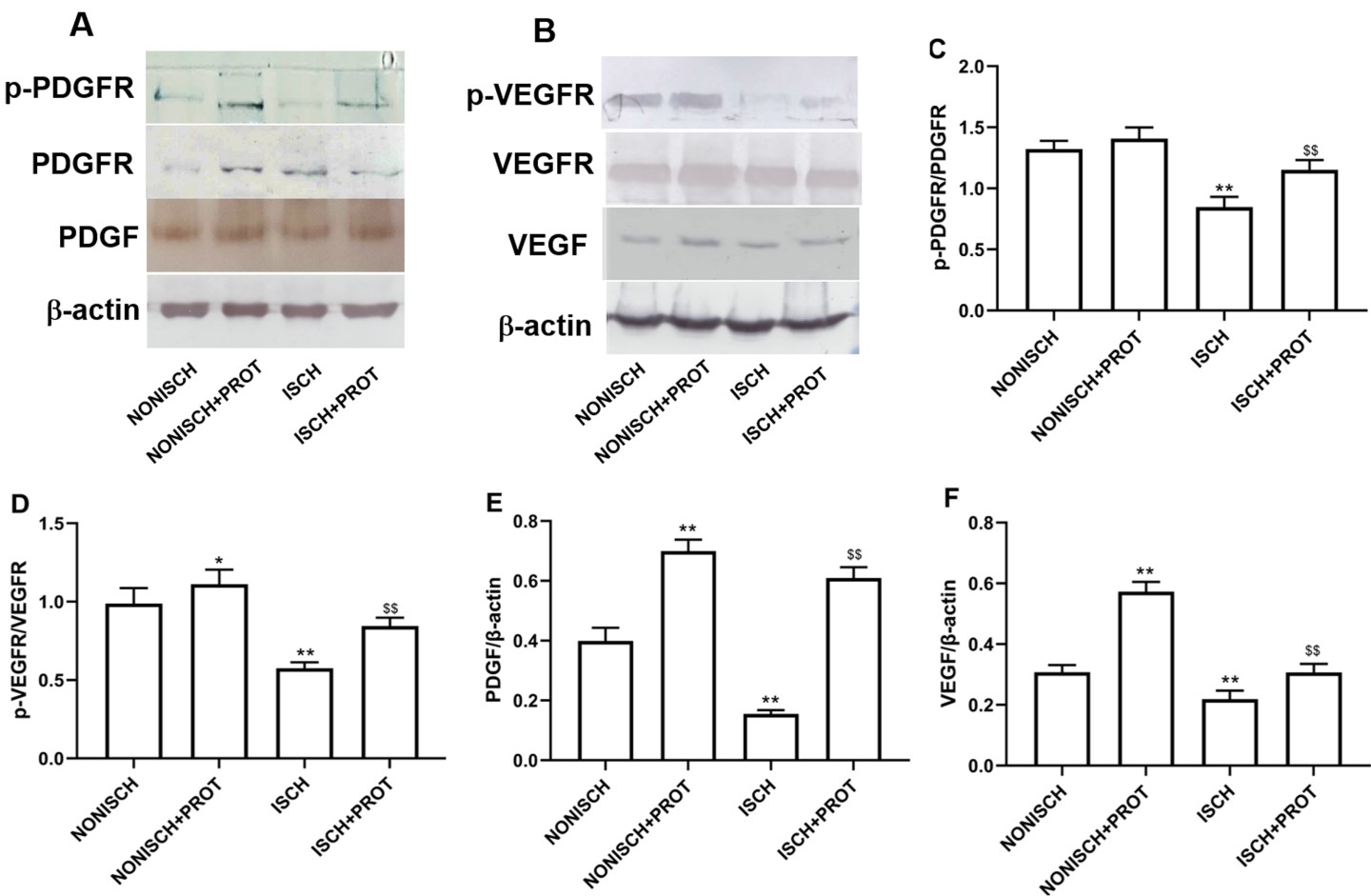

**Figure 6 Effect of the fusion protein on PDGF and VEGF signaling pathways.** Bar charts displaying relative protein levels. Data represent as means ± SD ($n = 6$). (A) Western blot showing PDGF signaling proteins. (B) Western blot showing VEGF signaling proteins. (C) Relative levels of p-PDGFR; (D) Relative levels of p-VEGFR. (E) Relative levels of PDGF. (F) Relative levels of VEGF. $*P < 0.05$, $**P < 0.01$ vs NONISCH; $^{\$\$}P < 0.01$ vs ISCH.

## Effects of the fusion protein on NF-κB pathway

NF-κB is a pivotal transcriptional factor to modulate inflammatory response. Inflammation is involved in wound healing and plays a critical role in the recovery mechanism. In present study, proteins related to NF-κB pathway were determined in wound using Western blot. Expressions of p-NF-κB and p-IKBα were significantly elevated in the ISCH group compared with the NONISCH group ($P < 0.05$) (Figs. 7A–7D). The fusion protein treatment reduced expression of p-NF-κB and p-IKBα in the ISCH +PROT group compared with the ISCH group ($P < 0.05$) (Figs. 7A–7D).

## DISCUSSION

In this study, our results suggest that the fusion protein accelerated ischemic wound healing and attenuated impairment of ischemic adductor muscle. Furthermore, the fusion protein enhanced the expression of the proangiogenic factors VEGF and PDGF, and levels of phosphorylated VEGFR and PDGFR. The fusion protein treatment was also observed

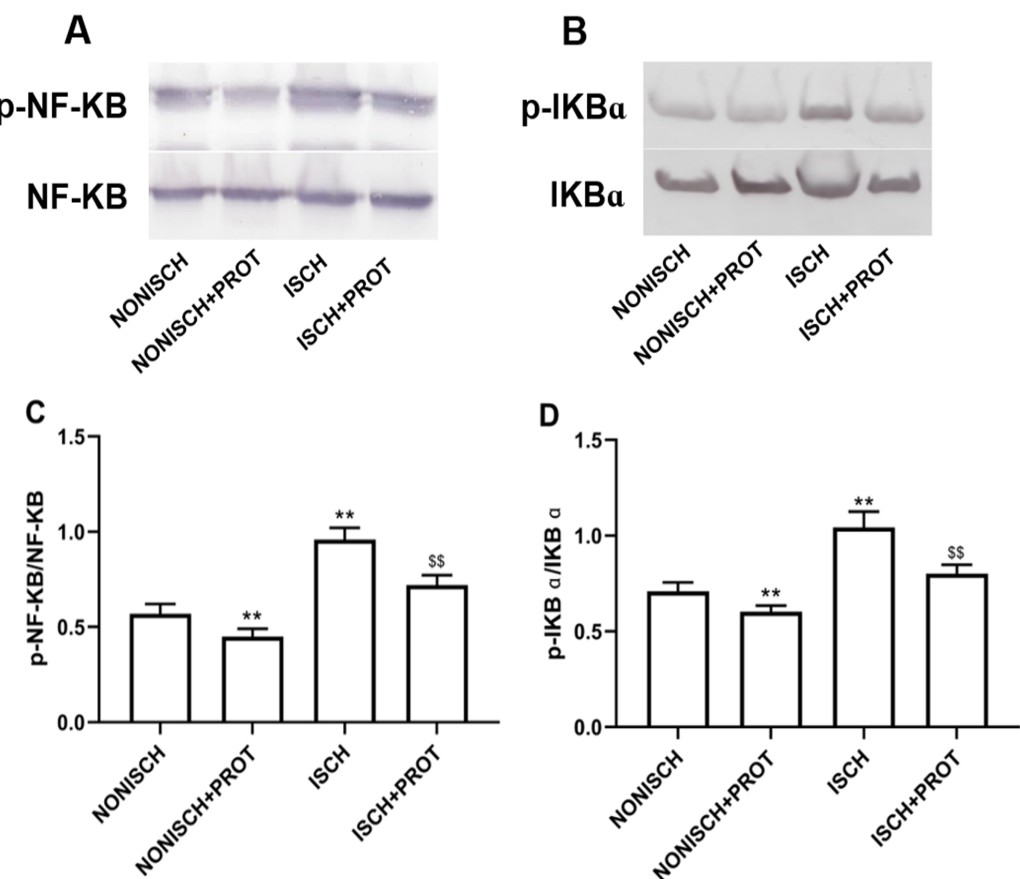

**Figure 7 Effect of the fusion protein on inflammation pathway.** Bar charts displaying relative protein quantification. Data represent as means ± SD ($n = 6$). (A) Western blot showing p- NF-KB . (B) Western blot showing p-IKB α. (C) Relative levels of p- NF-KB. (D) Relative levels of p-IKB α. **$P < 0.01$ vs NONISCH; $^{\$\$}P < 0.01$ vs ISCH.

to decrease levels of TNF-α and IL-6, and expression of p-NF-κB and p-IKBα. These findings suggest that the fusion protein exhibited its beneficial effects via improvement of angiogenesis and modulatory effects on inflammation.

Wound healing is a complex pathophysiological process, in which various precise mechanisms including the simultaneous actions of numerous cell types and modulators are required. The impairment of wound healing due to diabetes can be attributed to ischemia, so diabetic foot ulcers have become among the most dreaded complication of diabetes because of the high risk of amputation (*Marrotte et al., 2010*). Ischemia aggravates diabetic macro-angiopathy and reduces re-vascularization, postponing the development of granulation tissue and re-endothelialization (*Blakytny & Jude, 2006*). Lysozyme and antimicrobial peptides are regarded as host defense molecules owing to their antimicrobial role, but they are also involved in host physiological functions including inflammation and wound healing (*Nakatsuji & Gallo, 2012*). Cathelicidin, an antimicrobial peptide, has been reported to possess antifungal and antiviral properties (*Tsai et al., 2014*). Further study has shown that it promotes cutaneous wound repair through enhancement of re-epithelialization (*Heilborn et al., 2003*). Therefore, we explored the effect of the fusion

protein on the healing of ischemic wounds in diabetic mice. In this study, we observed that damage in hind limb muscles was more severe in ischemic diabetic mice than those that had normal levels of oxygenation, with ischemia reducing re-endothelialization and wound healing. Treatment with the fusion protein enhanced the healing of diabetic ischemic and non-ischemic wounds. Morphological characteristics caused by diabetes and ischemia, such as thinner granulation tissue and a degraded capillary network, were reversed. In addition, the fusion protein reduced the expression of Caspase 3 in the wounds.

Angiogenesis increases the supply of oxygen and nutrients to wounds. Therefore, it is beneficial for the formation of granulation tissue and re-epithelialization in wounds, further promoting wound healing. Angiopathy and reduced revascularization caused by a sustained lack of oxygen and nutrients due to ischemia, play important roles in the lack of healing in diabetic wounds (*Gershater et al., 2009*). Various studies suggest that increased angiogenesis promotes diabetic wound closure and neo-granulation tissue formation. It is well-known that proangiogenic factors such as VEGF and PDGF contribute to angiogenesis by inhibiting apoptosis and promoting cell proliferation, motility and survival (*Roskoski, 2008*). A reduction in VEGF in diabetes impairs angiogenesis, resulting in ischemic arterial morbidities (*Tchaikovski et al., 2009*). Several studies have indicated that therapies with VEGF and PDGF increase vascularization in diabetic wounds and accelerate diabetic wound healing (*Tchaikovski et al., 2009*). Overexpression of VEGF is reported to improve angiogenesis and blood perfusion in the ischemic hind limbs of mice (*Li et al., 2015*), and treatment with a combination of AMD3100 and PDGF-BB promotes neovascularization in diabetic wounds, and accelerates wound closure (*Allen et al., 2014*). Here, our results demonstrate that expression of VEGF and PDGF was significantly reduced in the muscles of ischemic diabetic limbs compared with normally-oxygenated limbs. Ischemia also increased the number of apoptotic cells. The fusion protein increased the expression of VEGF and PDGF.

Proangiogenic factors are involved in angiogenesis and accelerate wound healing, but other studies have suggested that impairment of their signaling rather than their expression reduces angiogenesis and delays diabetic wound healing (*Geraldes et al., 2009*). Therefore, enhancing the signaling related to angiogenesis may be beneficial to diabetic wound healing. Diabetes disrupts VEGF and PDGF signaling via reduced activation of VEGF and PDGF receptors, Akt and ERK, *etc* (*Lizotte et al., 2013*). HIF-1$\alpha$ is a vital cytokine which regulates angiogenesis in hypoxic tissue. Treatment with HIF-1$\alpha$ increases vessel density and improves limb perfusion (*Sarkar et al., 2009*). Our results indicate that expression of VEGF, PDGF and eNOS mRNA in ischemic limbs decreased in diabetic mice. Treatment with the fusion protein restored expression of these growth factors, enhanced phosphorylation of VEGFR and PDGFR, and increased capillary density in ischemic limbs. Treatment with the fusion protein also increased expression of HIF-1$\alpha$. Similarly, the number of capillaries in the granulation tissue also increased. Therefore, these results suggest that the fusion protein accelerated wound healing via increased VEGF and PDGF signaling.

Inflammation is a vital factor in cutaneous wound closure, and gradually subsides in the next five days (*Eming, Krieg & Davidson, 2007*). Hyperglycemia causes oxidative stress through diverse mechanisms such as glucose auto-oxidation, the polyol and hexosamine pathways and advanced glycosylation end products (*Evans et al., 2003*; *Nishikawa et al., 2000*). Oxidative stress triggers persistence of inflammatory response, inflammation is one of the most important mechanisms in diabetic complications including the delayed wound healing (*Soneja, Drews & Malinski, 2005*). It has been reported that excessive generation of proinflammatory cytokines such as TNF-α and IL-6 delays diabetic wound healing (*Fahey et al., 1991*; *Guo et al., 2007*; *Haidara et al., 2009*). Nuclear factor kappa B (NF-κB) is a vital transcription factor that plays an important role in various biological processes such as apoptosis, migration, cell proliferation and inflammation (*Ghosh & Karin, 2002*; *Hoesel & Schmid, 2013*; *Monkkonen & Debnath, 2018*). Furthermore, NF-κB has been reported to be involved in the pathogenesis of several chronic diseases (*Sanchez & Sharma, 2009*; *Wullaert, Bonnet & Pasparakis, 2011*). IKBα regulates activation of NF-κB via phosphorylation and degradation, phosphorylated IKBα activates NF-κB. Activated NF-κB translocates to the nucleus, binds to DNA and accelerates the transcription of proinflammatory cytokines and enzymes including IL-6, TNF-α, and iNOS (*Kim et al., 2010*; *Li et al., 2010*; *Tak & Firestein, 2001*). Sustained hyperglycemia is confirmed to result in abnormal activation of NF-κB in type 1 diabetes, which contributes inflammatory response and oxidative stress (*Giacco & Brownlee, 2010*). Pharmacologic research suggests that inhibited NF-κB ameliorates diabetic vascular function (*Kassan et al., 2013*). Activated NF-κB has reported to be involved in diabetic complications in various cell types (*Suryavanshi & Kulkarni, 2017*). Recent study has demonstrated that NF-κB/IKBα signaling pathway is upregulated in diabetic wound, and downregulation of NF-κB/IKBα signaling pathway improves diabetic wound healing (*Yuan, Das & Li, 2018*). Our data show that the fusion protein reduced levels of TNF-α and IL-6, downregulated NF-κB/IKBα signaling pathway in ischemic diabetic wound. These exhibit anti-inflammation of the fusion protein in ischemic diabetic wound.

## CONCLUSIONS

In summary, the present study indicates that the fusion protein promotes ischemic diabetic wound closure, alleviates damage of ischemic adductor muscles in diabetic mice. These positive effects are associated with acceleration of angiogenesis and decrease of inflammation. Therefore, the fusion protein may be an effective therapeutic drug against diabetic foot ulcers.

## ABBREVIATIONS

| | |
|---|---|
| **BCA** | bicinchoninic acid |
| **DAB** | 3,3'-diaminobenzidine |
| **eNOS** | endothelial nitric oxide synthase |
| **IKBα** | inhibitor of NF kappa B alpha |
| **FGF** | fibroblast growth factor |

| | |
|---|---|
| **HIF** | hypoxia inducible factor |
| **IL-6** | interleukin-6 |
| **NF-κB** | nuclear factor-kappa B |
| **PBS** | phosphate buffered saline |
| **PDGF** | platelet derived growth factor |
| **PDGFR** | PDGF receptor |
| **TNF-α** | tumor necrosis factor-alpha |
| **VEGF** | vascular endothelial growth factor |
| **VEGFR** | VEGF receptor |

### Funding

This study was supported by grants from the Army Key Project (No. BJN14C001), the National Natural Science Foundation of China (Nos. 811-72790, and 81671586), the Academic and Technical Leaders of Wannan Medical College (No. 010202041703), and University Outstanding Young Talents Project of Anhui Province (No. gxyq2017036). The funders had no role in study design, data collection and analysis, decision to publish, or preparation of the manuscript.

### Grant Disclosures

The following grant information was disclosed by the authors:
Army Key Project: BJN14C001.
National Natural Science Foundation of China: 811-72790 and 81671586.
Academic and Technical Leaders of Wannan Medical College: 010202041703.
University Outstanding Young Talents Project of Anhui Province: gxyq2017036.

### Competing Interests

The authors declare that they have no competing interests.

### Author Contributions

- Wei Li conceived and designed the experiments, performed the experiments, prepared figures and/or tables, and approved the final draft.
- Yu-Xin Jiang conceived and designed the experiments, performed the experiments, authored or reviewed drafts of the paper, and approved the final draft.
- Qing-Ying Chen conceived and designed the experiments, analyzed the data, authored or reviewed drafts of the paper, provide the fusion protein, and approved the final draft.
- Guo-Guang Wang conceived and designed the experiments, analyzed the data, prepared figures and/or tables, authored or reviewed drafts of the paper, and approved the final draft.

## Animal Ethics

The following information was supplied relating to ethical approvals (i.e., approving body and any reference numbers):

Animal Experimental Ethics Committee of Wannan Medical College approved this research (20180311).

## Data Availability

The raw measurements are available in the Supplementary Files.

## Supplemental Information

Supplemental information for this article can be found online at http://dx.doi.org/10.7717/peerj.11256#supplemental-information.

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
