# Peer review of "Recombinant fusion protein by lysozyme and antibacterial peptide enhances ischemic wound healing via angiogenesis and reduction of inflammation in diabetic db/db mice"

_PeerJ, doi:10.7717/peerj.11256_

## Round 0.1 · original submission · Major Revisions

Please address carefully the Reviewers comments.

Reviewer 1 ·

Basic reporting

All is OK except some images and figures which are not convincing

Experimental design

The high technical standard is not reached in some experiments like western blots

Validity of the findings

see my general comments to the author

Additional comments

The article “Recombinant fusion protein by lysozyme and antibacterial peptide enhances ischemic wound healing via angiogenesis and reduction of inflammation in diabetic db/db mice » is a well written paper in a good english with the aim to answer the question if the fusion protein « lysozyme and antibacterial peptide » could enchance the wound healing in diabetic mices via angiogenesis and reduction of inflammation. I have two main critics :
1- 1- The experiments were carried out with appropriate techniques and the results are convincing except for a few images and western blots like those in fig. 4, 6B and 7B; yet the statistical analyzes seem convincing.
2- 2- The second remark concerns the use only of diabetic mice in this work whereas the authors should have compared diabetic mice to non-diabetics, even if they would have been forced to reduce their investigations either on angiogenesis, or on reducing inflammation.

Other minor remarks:
Line 66- 67 preventing infection from pathogenic bacteria and viruses within wounds and fractures in trauma patients (Callewaert&Michiels 2010).>> This reference didn’t speak about Virus!

Line 112 NONISCH+PROT and ISCH+PROT mice were injected with fusion protein (0.1 mL, 50 μg/mL) >> You didn’t show or verify if the fusion is still active?; What about the purification of this enzyme and why did you use only one concentration ?
Line 123 Wound samples were homogenized in cold PBS s and centrifuged at 10000rpm for 20min at 4°C.>>please remove the “s”
Lines 152-153 2 mmol/L phenyl methylsulfonyl fluoride and protease inhibitors (10 μg/L leupeptin and aprotinin).>> the phenyl methylsulfonyl fluoride (PMSF) is also a protease inhibitor!

Line 168 infiltrationt and increase of the number >>please remove the “t” at the end of infiltration
Line 188 The fusion protein also elevated the levels of PDGF and VEGF in serum (P<0.05) (Figs. 3A and 3B)>> in the legend, it is said p<0.01
Line 198 enhanced expression of PDGF and CD34 in ischemic adductor muscle compared to the ISCH group (Figs. 4C and 4D) >> in the legend, it is said the inverse: (C) Representative images of adductor muscle sections stained for CD34 (x400). (D) Representative images of adductor muscle sections stained for PDGF (x400)

·

Basic reporting

Manuscript Number: 53150v1
Manuscript title: Recombinant fusion protein by lysozyme and antibacterial peptide enhances ischemic wound healing via angiogenesis and reduction of inflammation in diabetic db/db mice

This manuscript written by Wei Li et.al, to demonstrated the effect of recombinant fusion protein on wound healing in diabetic conditions. They have conducted a number of experiments and collected much valuable data for such a paper. The paper was written very well and the results are interpreted well in the aspect of diabetic wound healing. This manuscript is suitable for publication in this journal after minor revision of the manuscript based on the following queries.

1. The manuscript has issues with grammatical mistakes and should be corrected before publication.
2. Add the detailed methodology of fusion construction in materials and methods section. Whether you construct this fusion protein used in this manuscript? Or you obtained this fusion protein from any commercial manufactures?, please add this details.
3. again add some details about how many days you treated the animal with fusion protein, please exactly mention treatment period in materials and methods section.
4. How the authors picked the concentrations of fusion proteins in their experiments – Justify.
5. add the wound morphology image of control and all treated animal groups with wound area closure calculations.

Experimental design

1. Add the detailed methodology of fusion construction in materials and methods section. Whether you construct this fusion protein used in this manuscript? Or you obtained this fusion protein from any commercial manufactures?, please add this details.

2. again add some details about how many days you treated the animal with fusion protein, please exactly mention treatment period in materials and methods section.

Validity of the findings

No comment

Additional comments

-

---

## Round 0.2 · Major Revisions

I really didn't like how the authors responded to the reviewers' comments. When authors respond to reviewers, if they want the paper to be accepted, they have to respond carefully to the suggestions requested. The rebuttal letter is quite difficult to understand... The English language used is really difficult to follow. Probably the authors underestimate this step.

Here are some examples:

When you will give a proper response to this suggestion "the use only of diabetic mice in this work whereas the authors should have compared diabetic mice to non-diabetics", insert a sentence in the manuscript.

I can't see a proper answer to this comment: "add the wound morphology image of control and all treated animal groups with wound area closure calculations."

The manuscript is well written (much better than the rebuttal letter!!!), but some grammatical mistakes, as suggested by reviewers, are still present.

Check carefully that all the methods have been inserted: where is the method related to NF-kB western blot analyses? Also, please, write it properly the acronym in the paper.

Change Western blot figures you inserted in the paper with better figures that represent the results you discussed. Also, insert in each lane the name of the samples.

In figure 1E, insert the mean values and SD with statistical analysis.
If the RT-PCR analyses have been calculated with the 2-DDct method, the control sample should be 1. How these results have been calculated? Check all the figures, and make corrections.

This is your last chance to make properly the revision requested.

---

## Round 0.3 · accepted · Accept

The revision has been properly performed.

Reviewer 1 ·

Basic reporting

No comment

Experimental design

no comment

Validity of the findings

no comment

Additional comments

I found that the authors have answered all my questions and remarks; I thank them for their efforts